# Molecular fingerprints resolve affinities of Rhynie chert organic fossils

C. C. Loron ®[1] ✉, E. Rodriguez Dzul ®[1], P. J. Orr[2], A. V. Gromov[3], N. C. Fraser ®[4] & S. McMahon ®[1,5] ✉

The affinities of extinct organisms are often difficult to resolve using morphological data alone. Chemical analysis of carbonaceous specimens can complement traditional approaches, but the search for taxon-specific signals in ancient, thermally altered organic matter is challenging and controversial, partly because suitable positive controls are lacking. Here, we show that non-destructive Fourier Transform Infrared Spectroscopy (FTIR) resolves in-situ molecular fingerprints in the famous 407 Ma Rhynie chert fossil assemblage of Aberdeenshire, Scotland, an important early terrestrial Lagerstätte. Remarkably, unsupervised clustering methods (principal components analysis and K-mean) separate the fossil spectra naturally into eukaryotes and prokaryotes (cyanobacteria). Additional multivariate statistics and machine-learning approaches also differentiate prokaryotes from eukaryotes, and discriminate eukaryotic tissue types, despite the overwhelming influence of silica. We find that these methods can clarify the affinities of morphologically ambiguous taxa; in the Rhynie chert for example, we show that the problematic "nematophytes" have a plant-like composition. Overall, we demonstrate that the famously exquisite preservation of cells, tissues and organisms in the Rhynie chert accompanies similarly impressive preservation of molecular information. These results provide a compelling positive control that validates the use of infrared spectroscopy to investigate the affinity of organic fossils in chert.

Vibrational methods such as Raman and Fourier Transform Infrared (FTIR) microspectroscopy are increasingly popular tools for the investigation of fossil organic matter[1]. These methods are widely available, rapid, and essentially non-destructive, and they can be applied to fossils in situ without removing the sedimentary matrix. Raman spectroscopy uses a monochromatic laser and relies on the Stokes Raman effect whereby incident photons are shifted to a lower frequency by an amount corresponding to the vibration of a specific chemical bond. FTIR spectroscopy uses a polychromatic infrared light source and a Michelsen interferometer; a Fourier transform of the resulting interferogram generates an intelligible transmission or reflectance spectrum with bands at specific wavenumbers, again

corresponding to specific chemical bonds. These methods can detect methyl, methylene, carbonyl, carboxyl, aromatic and other moieties in ancient organic fossilization products and thereby shed light both on the original composition of the precursor biomolecules and on the pathways of chemical alteration during and after fossilization[2,3].

Previously, Raman spectroscopy has been used to survey phylogenetic and physiological diversity of organically preserved Phanerozoic metazoans[4], to differentiate vertebrate from invertebrate tissues in the Carboniferous Mazon Creek Lagerstatte[5], to identify colour pigments in fossilized dinosaur eggs[6], to investigate the biogenicity of Precambrian microfossils[7] and to determine kerogen maturity (e.g., refs. [8–10]). Similarly, FTIR spectroscopy and chemometrics have been

[1]UK Centre for Astrobiology, School of Physics and Astronomy, University of Edinburgh, Edinburgh, UK. [2]UCD School of Earth Sciences, University College Dublin, Dublin, Ireland. [3]EastCHEM and School of Chemistry, University of Edinburgh, Edinburgh, UK. [4]Natural Sciences Department, National Museums Scotland, Edinburgh, UK. [5]School of Geosciences, University of Edinburgh, Edinburgh, UK. ✉e-mail: v1cloron@ed.ac.uk; sean.mcmahon@ed.ac.uk

applied in chemotaxonomy[11,12], the determination of kerogen maturity (see review in ref. [13]), the characterization of individual microfossils (e.g., refs. [2,14–17]) and the identification of bacterial, eukaryotic and archaeal lipid fossilization products in various Precambrian successions[3,18–20].

These studies suggest that vibrational microspectroscopy can reveal physiological and phylogenetic information about organic fossils independently of other lines of evidence; this would be especially useful in the study of fossil microbes, the morphology of which, even when well preserved, is often not diagnostic of their phylogenetic affinities. In principle, such evidence could elucidate the diversity of important organic Lagerstätten in a way that traditional palaeontological methods have been unable to achieve. An essential first step, however, is to confirm the relationships between the phylogenetic affinity and microspectroscopic fingerprints of ancient, thermally altered microfossils within a single assemblage.

To achieve this, the Lower Devonian (~407 Ma) Rhynie chert assemblage of Aberdeenshire, Scotland, famous for its exceptionally preserved fossils since they were discovered more than a century ago, provides ideal material. These fossils open a unique window on an early terrestrial biotic community and include some of the oldest known terrestrial arthropods, land plants, fungi, heterocystous cyanobacteria, and various other microbial eukaryotes and prokaryotes (see review in ref. [21]). They are composed of brown-black carbonaceous matter permineralized and encased by transparent microcrystalline quartz, which derives from silica precipitated by terrestrial hot springs[22]. As such, they can be examined in transmitted light under an optical microscope, revealing exquisite, micron-scale, morphological detail. To date, almost all studies of the Rhynie chert biota have focused on the identification and systematic description of new taxa using this approach[21,23]. Chemical investigations have been limited but promising[24–26]. FTIR spectroscopy has confirmed the presence of aliphatic CH bands and subordinate amide, C=O and COOH bands, but taxon-specific differences within the assemblage have not yet been reported[25,26].

In this work, we analyze 49 individual fossils from the Rhynie chert assemblage using attenuated total reflectance (ATR) FTIR microspectroscopy. The acquired infrared spectra are then compared using multivariate and machine-learning approaches. We show that differences between prokaryotes (cyanobacteria) and eukaryotes and between eukaryotic tissue types can be identified based on the fossilization products of lipids, sugar, and protein.

## Results

The fossils (Fig. 1 and Supplementary Fig. 1) are located at the surface of polished thin sections and represent nine individual fossil cyanobacteria (*Rhyniosarcina devonica*[27], *Palaeolyngbya kerpii*[28], and a multiseriate *Stigonema*-like taxon), six unclassified arthropod cuticles, four unclassified plant spores, eleven plant cortices (*Aglaophyton majus*, *Rhynia gwynne-vaughanii*) and one set of rhizoids, four peronosporomycetes (*Frankbaronia polyspora*), two specimens of the amoebozoan *Palaeoleptochlamys hassii*[23] and ten fungi (six *Palaeomyces* sp.; four glomeromycotan vesicles preserved within plants). We also analyzed two specimens of the enigmatic tubular nematophyte, *Nematoplexus rhyniensis*[29], which has been interpreted as either a (lichenized) fungus or a plant[29,30]. These individuals were chosen for their phylogenetic diversity, excellent preservation and lack of visible carbonization, and presence at the thin-section surface.

ATR-FTIR spectra exhibit typical characteristics of organic fossils preserved in chert (Supplementary Fig. 3). Absorption below 1400 cm⁻¹ is masked by intense absorption bands of silica (Si−O stretch). Seven absorption bands at ~1995, 1870, 1793, 1684, 1610, 1525, and 1492 cm⁻¹ are present in every specimen and are associated with Si−O overtones in quartz[19]. Broad absorption between 3800 and 3000 cm⁻¹ is attributed to molecular $H_2O$ in the chert matrix, various OH groups, and NH

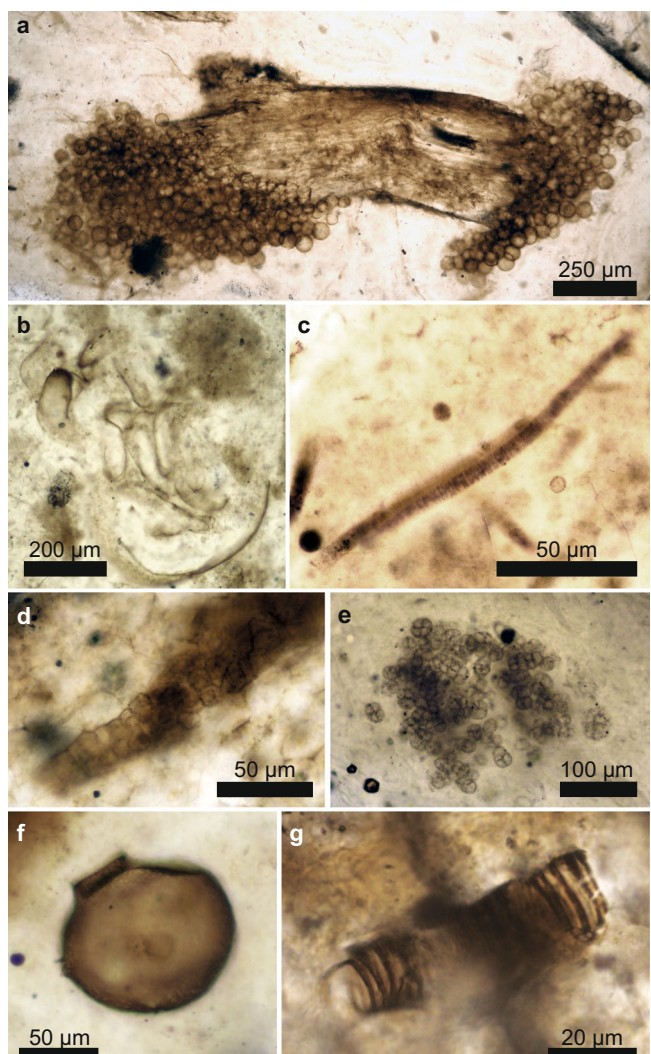

**Fig. 1 | Rhynie chert fossils.** Photomicrographs of representative Rhynie chert eukaryotic (**a**, **b**, **f**, **g**) and prokaryotic (**c**–**e**) groups of organisms. *Palaeomyces* sp. (fungi) on plant stalk (**a**); Arthropod (**b**); *Palaeolyngbya kerpii* (cyanobacterium, **c**); Stigonema-like cyanobacteria (**d**); *Rhyniosarcina devonica* (cyanobacterium, **e**); *Palaeoleptochlamys hassii* (putative arcellinid amoeba, **f**); *Nematoplexus rhyniensis* (nematophyte, **g**). Photomicrographs were taken by ER, SM, and CCL at the University of Edinburgh.

stretching vibrations[31]. In addition, absorption bands of organic matter are recognized in the interval 3000–2800 cm⁻¹ and 1800–1400 cm⁻¹ and are interpreted as characteristic absorptions for different CH, C=O, COOH, and nitrogen-moieties (see Supplementary Table 1 and Supplementary Note). These organic groups represent the fossilization products of biomass that was originally dominated by lipids, proteins, and sugars. Long-chained aliphatic lipids are known to be very resistant to degradation and prone to fossilization[32,33]. Although less resistant to diagenetic degradation, protein, and sugar are modified via oxidative crosslinking forming carbonyl and carboxyl rich peroxidation products as well as N- and S-heterocyclic polymers[5,33].

After preprocessing (see Methods), spectra were submitted to two unsupervised methods of clustering: K-mean clustering and principal component analysis (PCA). Both methods naturally sorted the data between eukaryotes and prokaryotes, i.e., cyanobacteria (Fig. 2). In the PCA, these domains are separated on PC1 (49% variance); the loading (Supplementary Fig. 5) shows that the separation is due chiefly to variation in the CH region corresponding to differences in lipid composition. PC2 (25% variance) loading corresponds to fossilization

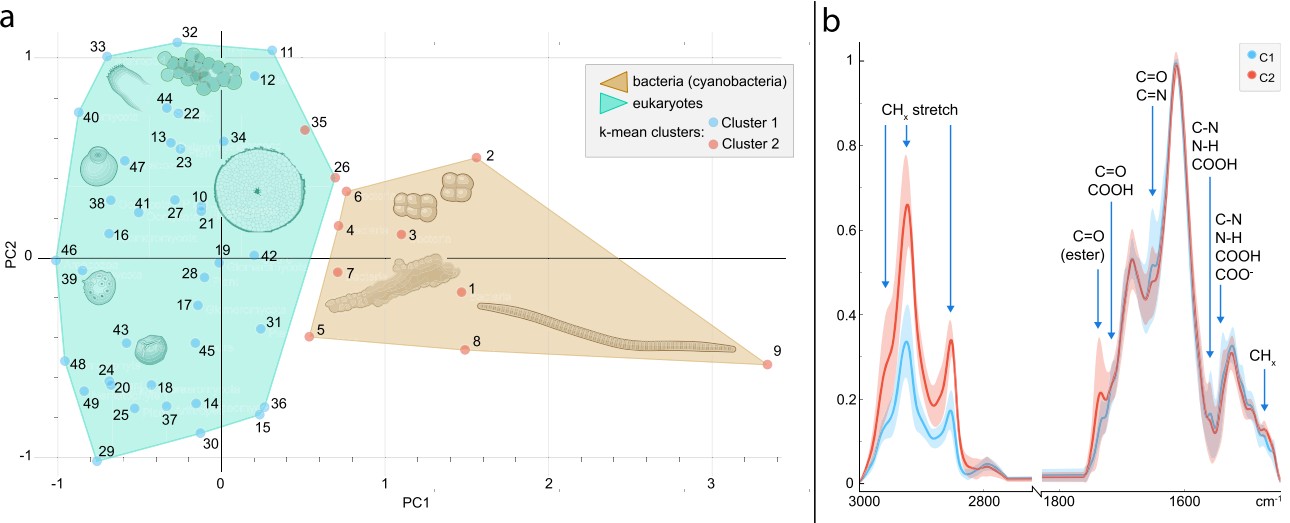

**Fig. 2 | Unsupervised analysis of Rhynie chert fossils. a** K-means clustering (blue/red dots) and score plot for Principal Component Analysis (PCA) separating eukaryotes from prokaryotes on PC1 (convex hulls). Numbers correspond to the specimen numbers in Supplementary Data 1. The cartoons represent an arthropod fragment, a plant spore, a plant axis, *Palaeoleptochlamys hassii* (putative arcellinid amoeba), *Frankbaronia polyspora* (putative peronosporomycete), and three cyanobacteria: *Palaeolyngbya kerpii*, *Rhyniosarcina devonica*, and an unidentified *Stigonema*-like taxon. Cartoons were drawn from Rhynie chert organism photographs taken by ER, SM, and CCL. Source data are provided in the Source Data file. **b** Average spectra from k-mean clustering showing the differentiation between the groups. Organic bands correspond to absorption of lipids (bands numbers 1–3, 9 in Supplementary Table 1) and fossilization products of sugar and protein (bands 4–8); see Supplementary Note. Red and blue area represent the full variability of spectra between the specimens of each group (C1 and C2). Thick lines in each area represent average spectra for each group (mean values of each intensity at each wavenumber).

products of sugar and protein (C=O, COOH, N-moities) but is less informative to sort the data, probably due to heterogeneities prior to permineralization (e.g., microbial degradation of plant pieces). K-mean clustering suggested two clusters corresponding to a similar sorting to PC1 with 96% accuracy (only 2 eukaryotes (numbers 26 and 35) were sorted as prokaryotes; Fig. 2a, b).

Subsequently, spectra were divided into two classes, eukaryotes and prokaryotes (based on their reported affinity). Nematophytes were excluded from the model. They were then submitted to supervised machine learning algorithms (K-nearest neighbour, logistic regression, and Random Forest). The K-nearest neighbour algorithm correctly classified all spectra from the test set, validating the approach to discriminate between domains using infrared data. Inclusion of the nematophyte data in the model predicted a eukaryotic affinity for these fossils.

The chemometric ratio $R_{3/2}$, i.e., the ratio of the peak intensities corresponding to $CH_3$ and $CH_2$, has been proposed as a domain indicator of bacteria, archaea and possibly eukaryotes[3] because membrane lipids in archaea are characterized by shorter and more highly branched aliphatic chains than those of bacteria and eukaryotes[3,18,19]. However, discriminating the three domains in natural assemblages based only on this ratio may be difficult[34]. Our specimens yield $R_{3/2}$ values between 0.25 and 0.63, in accordance with values expected for non-fossil bacterial and eukaryotic lipids. However, the measured ratios do not discriminate between prokaryotes (average 0.43; $\sigma\bar{x} = 0.03$) and eukaryotes (0.42; $\sigma\bar{x} = 0.01$), unlike in living cells[3]. Mann–Whitney $U$ tests did not reveal significant differences between the ratios obtained for prokaryotes and eukaryotes ($p$-value = 0.76652). As suggested by Igisu et al. (2012)[34], $R_{3/2}$ cannot be applied unequivocally as a domain discriminant (eukaryote vs prokaryote) to all organic fossils. Possibly the Rhynie chert is unusual in this respect; other, more mature, fossil assemblages may be more dominated by the fossilization products of lipids, whereas eukaryotes in the Rhynie chert contain a range of fossilization products that can influence the $R_{3/2}$ ratio[3]. Alternatively, another Rhynie chert study suggests that prokaryotes and eukaryotes may be

discriminated when Raman I-1350/1600 ratio (degree of structural order) is plotted against $R_{3/2}$ ratios[26].

The intensity ratio of carbonyl ester and methylene (ester/$CH_2$) has also been proposed to discriminate different living bacteria and archaea based on the main differences in their lipids[35]. This method has been successful in distinguishing bacteria with ester-linked lipids (*E. coli, B. subtilis, M. lysodeikticus*) from bacteria with ester and ether lipids (*T. commune*)[35]. Archaeal cell membranes are constituted of glycerol-ether lipids, whereas bacterial and eukaryotic membranes consist mainly of glycerol-ester lipids[36]. A variable contribution of carbonyl ester groups is observed in fossils; thus this ratio might constitute a chemometric proxy for domain discrimination. Mann–Whitney $U$ tests showed significant differences in ester/$CH_2$ ratio between prokaryotes and eukaryotes from the Rhynie material ($p$-value = 0.02). The eukaryotes as a whole show higher values than prokaryotes, consistent with the former being richer in ester-linked lipids[36]. However, the large variation between eukaryote groups (Supplementary Fig. 2) suggests that the ester/$CH_2$ ratio alone may not reliably discriminate between fossil domains.

We tested alternative ratios, C=O/$CH_2$ (carbonyl/$CH_2$) and N/$CH_2$ (Nitrogen-moieties/$CH_2$), corresponding to the variation in contribution of products from the fossilization of sugars and proteins. These ratios differentiate the two domains in Rhynie material (Mann–Whitney $U$ test, $p$-value = 0.00002 (C=O/$CH_2$) and 0.003 (N/$CH_2$)). Higher values for eukaryotes are consistent with the result of multivariate analyses showing a lower contribution of lipids to their general composition (therefore a lower $CH_2$ contribution). Products from the fossilization of saccharide-rich tissues would contain various amounts of carboxyl (COOH), carbonyl (C=O), and N-moieties[5,37] and, in the cell wall, preservation of carbohydrate and protein material can be favoured by the reconformation of long-chained aliphatic moieties[38]. Sugars and proteins in bacterial extracellular polymeric substances (EPS) and cell walls may be more prone to degradation, while the esterification of carboxyl-rich EPS at the silica-kerogen interface, together with existing EPS esters[39,40], may explain the high absorption from ester C=O in the bacterial spectra (Fig. 2B). Taken

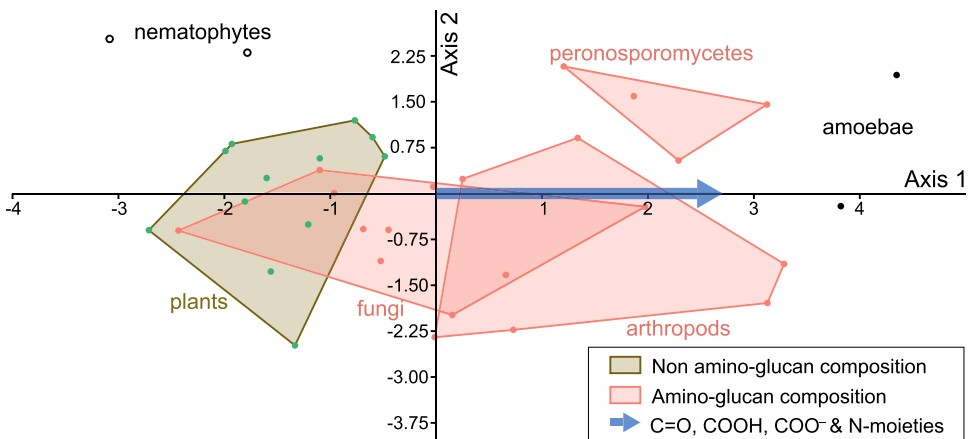

**Fig. 3 | Canonical Correspondence Analysis of Rhynie chert fossils.** Specimens are distributed according to the organic bands relative to fossilization products of sugars and protein (carbonyl, carboxyl, and N-moieties), showing rightward an enrichment in N-moieties (axis 1). See Supplementary Figure 9 for detail on the specific bands responsible. Source data are provided in the Source Data file.

together, ester/CH$_2$, C=O/CH$_2$, and N/CH$_2$ chemometric ratios constitute useful tools to narrow down the origins of fossil material in the Rhynie chert.

To understand how variations in sugar-protein fossilization products might discriminate between the eukaryotes, we adapted for ATR-FTIR data the methods used for Raman data by refs. [4,6,41] and [5]. We transformed the intensity of informative organic bands (as revealed by the multivariate analyses) for each specimen (excluding the plant spores) into a matrix and conducted a discriminant analysis (Fig. 3). Living opisthokont biomass (fungi and animals) is rich in amino-sugar (chitin, chitosan) but fungi are also rich in β-glucan[42]. Modern Peronosporomycota, although containing cellulose, have a cell wall with up to 10% of N-acetylglucosamine-based carbohydrates[43]. Plant cells are richer in non-amino-sugars[44]. On the other hand, plant spores are composed of resistant aromatic polymers termed sporopollenin[45]. Discriminant factors along the first axis correspond to fossilization products of sugar and protein (C=O, COOH and N-moities). The rightward distribution of chitin-bearing organisms on the first axis may be explained by a higher contribution of amino-compounds from the degradation of chitinous tissues. The position of *Palaeoleptochlamys hassii* suggests a composition enriched in amine-containing fossilization products. Strullu-Derrien et al.[23] pointed out an organic (non-agglutinated) composition for the test of this taxon and interpreted it as an arcellinid amoebozoan (testate amoeba). The ATR-FTIR results are consistent with an organic arcellinid affinity[46]. For the enigmatic tubular nematophyte, *Nematoplexus*, our FTIR data support a eukaryotic affinity, according to both unsupervised and machine-learning methods, suggesting a cyanobacterial-dominated lichen is the less likely option. Its position on the discriminant analysis (Fig. 3) indicates the composition is most similar to plant fossilization products (non-amino-sugars), excluding the possibility its composition was chitinous.

The remarkable quality of preservation in the Rhynie chert is well attested and extends to very fine cellular features, soft-tissues, and mutual associations between taxa in life positions (e.g., refs. [47,48]; see review in ref. [21]) all of which were facilitated by rapid silicification[23]. The excellent morphological preservation, together with the excellent chemical preservation reported here, suggest a low thermal maturity profile. A previous study[26] using Raman geothermometry has estimated the thermal maturity of Rhynie chert carbonaceous matter as between 190–310 and 245–345 °C, depending on the geothermometer used. However, the Raman spectra presented in that study are strikingly similar to the low thermal maturity spectrum presented by Lahfid et al.[8], suggesting a similar temperature (ca. 160 °C), more consistent

with the quality of fossil preservation and our FTIR results; in particular, the Raman spectra in Qu et al.[26] show strong bands for disordered carbon (D3, D4, D5[8–10]).

Although the fossils analyzed here evidently followed similar taphonomic and diagenetic pathways and are all extremely well preserved, some minor variation between them is expected, for example, in respect of the microenvironmental conditions in which they fossilized or the extent of microbial degradation (although only well-preserved tissues were included in the dataset). In addition, the thin sections were not all prepared at the same laboratory; the eukaryotes are in nine thin sections obtained from two different sample collections, and the cyanobacteria are in four thin sections obtained from three different sample collections. Nevertheless, the overall spectra are very similar, and taxon-specific signatures appear not to be confounded by any of these variables.

Organic matter in some ancient chert-hosted microfossils may have been replaced at a late stage by mobile hydrocarbons[49,50]. There is no doubt that the organic matter comprising Rhynie chert fossils is original; its colour and texture varies between tissue types and there is no evidence for late-stage replacement. We interpret the differences observed to be variations in the original precursors of the fossil organic matter, modified through diagenesis, rather than resulting from external contamination or preparation. Indeed, our results imply that taxon-specific microspectroscopic data could be used in future to determine the syngenicity, biogenicity and likely affinities of organically preserved structures in other cherts, as hydrocarbons of different composition are unlikely to replace different taxa selectively.

This study shows that the remarkable quality of morphological preservation characteristic of the ~407 Ma-old Rhynie chert fossil assemblage is accompanied by similarly impressive preservation of chemical information. Contrary to many older (i.e., Precambrian) cherts to which vibrational spectroscopic methods have been applied, in the Rhynie chert one can distinguish visually between prokaryotes and eukaryotes, often with well documented taxa[21], and thus confirm that the spectroscopic data map exceptionally well to phylogenetic affinity. These results therefore provide a positive control for future studies of more ambiguous material and have three additional implications. Firstly, the chemometric ratio R$_{3/2}$ is not an infallible guide to domain-level affinity (as already pointed out in a previous study[34]; here it does not significantly separate prokaryotes and eukaryotes even within a single, thermally immature fossil assemblage (although other ratios based on the occurrence of sugar and protein fossilization products do achieve this). Secondly, unsupervised methods can sort

FTIR spectra into eukaryotes and prokaryotes (without preselecting preferred absorption bands), while machine learning (supervised) methods offer the potential for predictive chemotaxonomy and the resolution of ambiguous fossil affinities (as in the case of *Nematoplexus*). Finally, we show that FTIR spectra obtained from chert-hosted fossils retain compositional specificity despite the overwhelming silica signal. Overall, these results open a unique window on the diversity of early terrestrial life while validating the use of ATR-FTIR spectroscopy, a rapid, widely available, non-destructive in situ method, for the resolution of biomolecular differences between ancient organically preserved fossils in chert.

## Methods

### Sample preparation

Samples were borrowed from the School of Geosciences, University of Aberdeen (hand samples), the Oxford University Museum of Natural History (polished thin sections), and the National Museum of Scotland (NMS, hand samples). Polished thin sections from the Aberdeen samples were prepared at University College Dublin. Polished thin sections from the NMS samples were prepared by Ivan Febbrari at the University of Edinburgh. Thin-section numbers corresponding to individual fossils are provided in Supplementary Data 1.

### Optical microscopy

Optical microscopy was conducted on a Leica DM2700P at the UK Centre for Astrobiology (University of Edinburgh). Photographs were acquired in transmitted light using ×10 and ×20 objectives and the Leica Application Suite (4.0) software.

### ATR-FTIR

Attenuated Total Reflectance-FTIR was conducted at room temperature on a Smiths IlluminateIR microscope equipped with liquid nitrogen cooled MCT detector providing spectral resolution of 4 cm$^{-1}$, using a diamond coated ATR objective (magn x36) at the School of Chemistry (University of Edinburgh). Backgrounds were taken in the air before analysis. Spectra were acquired in reflection mode for each specimen by combining 128 accumulations in the range 4000–650 cm$^{-1}$ with an aperture of 100 μm with the software Qual ID 2.51 (Smiths).

### Data processing

FTIR spectra were preprocessed before multivariate and machine learning analyses in the software Quasar 1.6.0 (Orange[51,52]). For analyses, spectra were truncated to 3000 and 1400 cm$^{-1}$ in order to remove the uninformative intervals due to OH absorptions (4000–3000 cm$^{-1}$) and intense vibration of silica (1400–650 cm$^{-1}$). Similarly, the interval between 2700 and 1760 cm$^{-1}$ was cut to remove the influence of atmospheric $CO_2$, ATR diamond absorption and high wavenumber silica overtones. Rubberband baseline correction and Min-Max normalization (see below) were applied to the spectra before analyses. For inspection and band assignment, average second derivative spectra for each domain (Eukaryote and Prokaryote) were obtained by applying a Savitzky–Golay filter of polynomial 2 with a window of 9 (Supplementary Fig. 4).

### Normalization.

To minimize the influence of thickness and differences in organic matter concentration, normalization of the data (spectra) is necessary[53]. We performed a min-max normalization by artificially setting the minimal intensity at 0 and the maximal intensity at 1 (corresponding to the 1615 cm$^{-1}$ silica band). The normalization was conducted in the preprocessing widget of the software Quasar. Such normalization results in the expression of all other absorption bands relative to the most intense band (in our case the ratio X/Silica).

Sanity checks were performed on the PCA results to ensure that the detected clustering was a result of compositional differences and not simply the changes in the abundance of organic matter in regard

to silica (presence of a larger fossil for example). In the latter case, the ratio organic/silica would follow broadly the same trend for each organic band. To test this, we calculated the ratios $CH_2$/silica and C=O/silica by dividing the intensity of the 2920 cm$^{-1}$ ($CH_2$) and 1650 cm$^{-1}$ (C=O) absorption bands by the 1615 cm$^{-1}$ Si–O band. Results were implemented on PCA score plots. We can see by comparing Supplementary Figures 6 and 7 that the increase of $CH_2$/silica ratio rightward (PC1) is not followed by the increase of the C=O/silica ratio; note also that the cyanobacteria, which might intuitively be suspected to preserve less organic material than plants, arthropods, etc., are actually characterized by higher $CH_2$/silica ratios, not lower. A similar observation can be made for PC2, along which C=O/silica increases but not $CH_2$/silica. In addition, the other bands of silica does not contribute clearly to the loadings (Supplementary Fig. 5). Loadings for PC1 and PC2 display features characteristic of lipid moieties and soft tissue fossilization products, respectively. The clustering of the different specimens is therefore compositional. Close inspection of Fig. 2B shows that the same is true of the k-means clustering: C1 is characterized by smaller CH stretching peaks but stronger peaks for several other organic moieties, so the primary difference between the two clusters is not simply the amount of organic matter.

### Data analyses

**Semi-quantitative ratios.** Calculations of methyl/methylene ratio ($R_{3/2}$) were conducted using the method introduced by[3]. After baselining of the spectra between ca. 3000 and 2800 cm$^{-1}$, intensity of $CH_3$ asymmetric stretch (2960 cm$^{-1}$) is divided by intensity of $CH_2$ asymmetric stretch (2925 cm$^{-1}$). Other parameters were obtained by dividing the intensity at 1650 cm$^{-1}$ (C=O); 1735 cm$^{-1}$ (C=O in ester group); 1560 cm$^{-1}$ (NH bending, CN stretching, COOH) by the intensity at 2925 cm$^{-1}$ ($CH_2$ asymmetric stretch). Statistical tests to test the difference between prokaryotes ($n = 9$) and eukaryotes ($n = 40$) for each of the ratio results were performed in PAST 4.09. A Shapiro–Wilk test shows that the data do not consistently follow a normal distribution. Therefore, non-parametric Mann–Whitney $U$ tests were applied to compare the two groups (eukaryotes and prokaryotes).

**Multivariate analyses.** Principal component analysis (PCA) and K-means clustering (KMC) were performed in Quasar. KMC was performed with random initialization and 300 iterations. PCA was performed on 6 components, explaining 96% of the variance. Only PC1 (49%) and PC2 (25%) were considered informative. Loadings for PC1 and PC2 are available in Supplementary Figure 5. Additional PCA include all fossils and 4 matrix samples is shown of Supplementary Fig. 8. Discriminant analysis was performed in PAST 4.09 using absorption band intensities at ca. 1460, 1540, 1560, 1650, 1715, 1735, 2851, 2920, 2960 cm$^{-1}$.

**Supervised analyses (machine learning).** Classification and predictions were done using Quasar. Preprocessed spectra were divided into a training set containing 60% of the spectra (29) and a test set containing 40% of the spectra (18). In order to build a consistent model, the training set was submitted to three different learners: (1) logistic regression with L2 ridge regularization (C=1); (2) k-nearest neighbour (kNN) using Euclidean distances and 5 neighbours; and (3) Random Forest (RF) with 50 trees. The three methods were applied with a leave-one-out cross validation. The classification accuracy on the training set was 97% for Random Forest with a ROC-AUC (Receiving Operator Characteristic Area Under the Curve) of 0.98. Logistic regression and kNN both provided 90% classification accuracy and a ROC-AUC of 0.98 and 0.97, respectively. We then used the three models to predict the classes within the test set. The classification accuracies were 94% for Random Forest (ROC-AUC 0.9) and Logistic Regression (ROC-AUC 1) and 100% for kNN (ROC-AUC 1).

# Article

**Reporting summary**

Further information on research design is available in the Nature Portfolio Reporting Summary linked to this article.

## Data availability

Thin-sections are available from their respective collections: Specimen codes: 2829p: Oxford University Museum of Natural History. Rfa, Rfe, Rff, Rfg, ALG2, AGLyon 2019, AGL127, Agl75 2019-1: University of Aberdeen. NMSRC9: National Museums of Scotland. Raw spectra are available on Edinburgh DataShare at https://doi.org/10.7488/ds/3806. Source data are provided as a Source Data file. Source data are provided with this paper.

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

## Acknowledgements

This research was supported by The Royal Society (UK) and the Belgium Wallonia Brussels programme WBI.WORLD (C.C.L.) and the CONACYT (National Council of Science and Technology of Mexico) Postgraduate Scholarship for Studying Abroad (E.R). We thank J. Parnell at the University of Aberdeen, the National Museums Scotland and the University of Oxford Museum of Natural History for facilitating access to samples. I. Febbrari is thanked for the preparation of thin sections. S. Hetherington and M. Krings are thanked for their help identifying cyanobacterial and plant taxa. D. Marosi-McMahon is thanked for the help with statistical comparison tests.

## Author contributions

C.C.L., S.M., and P.J.O. conceived the study. S.M., N.C.F., and P.J.O. acquired and prepared samples. S.M., E.R., and C.C.L. acquired photomicrographs. C.C.L. and A.V.G. acquired the ATR data. C.C.L. processed the data and conducted multivariate and supervised analyses with input from S.M. and E.R. C.C.L., and S.M. drafted the manuscript and figures. All authors reviewed and edited the manuscript.

## Competing interests

The authors declare no competing interests.
