## [Peer Review File · Nature Communications]

Molecular fingerprints resolve affinities of Rhynie chert organic fossilsReviewers' Comments:

Reviewer #1:

Remarks to the Author:

This paper presents results of ATR FTIR microspectroscopy performed on organic microfossils preserved in Lower Devonian Rhynie Chert in order to investigate the relationships between the phylogenetic affinity and microspectroscopic fingerprints of microfossils within a single assemblage. Evaluating affinities of fossil microorganisms is very important for elucidation of their origin, but only morphological characteristics is often insufficient. While chemical analysis of microfossils is promising to provide broad phylogenetic information, understand of taxon-specific chemical signals in microfossils is still limited. The authors analyzed 49 individual fossils (morphologically different 13 taxa) by ATR-FTIR and Raman microspectroscopies together with statistical analysis. Finally, the authors revealed that ATR-FTIR with statistical analysis can discriminate the prokaryotic (cyanobacterial) and eukaryotic fossils.

The present ATR-FTIR work of organic microfossils is very interesting. The authors have made a good attempt at analyzing microfossils by using ATR-FTIR. Because ATR-FTIR microspectroscopy can be applied to standard petrographic thin sections (removing from a glass slide is not needed), this can be widely applicable to geological samples. Microspectroscopy with statistical analysis helps to understand slight differences in obtained spectra. ATR-FTIR microspectroscopy together with statistical analysis could provide important insights for evaluating taxon-specific chemical signals in microfossils, however, still have some problems as indicated below.

Major comments:

(1) Peak assignments of IR spectra (Supplementary material Line 52-54)

I am wondering about the peak assignments for the analyzed samples. As far as I can see the raw spectra (Supplementary figure 3), the obtained peaks (the authors assigned as organic bands) below the 1900 cm^{-1} region are very small. Therefore, it is questionable whether the observed peaks in figure 2 and supplementary figures is due to fossil organic matter, due to matrix, or due to surface or experimental contamination.

More careful discussion would be needed for the peak assignments for the analyzed samples. At least, an IR spectrum of chert matrix should be added in main figure or supplementary material.

(2) A meaning of a ratio X/silica

In supplementary material line 73-75, the authors mentioned the spectra were normalized by the 1615 cm^{-1} silica band. This corresponds to a ratio X/Silica. In this case, CH/Si-O and other ratios depend on concentration/amount of organic matter in analyzed area. In many organic microfossils, they have hollow interior (less organic matter) and we can recognize them by wall/membrane structures. When such microfossil cell aggregates (e.g., Figure1) were analyzed by FTIR with 100 x 100 μm^2 aperture, the aggregates with smaller cells (i.e. prokaryotes) would be expected to show larger CH/Si-O than those with larger cells (i.e. eukaryotes). Therefore, I am wondering whether the PC1 in Figure2 depends on morphological parameter (e.g. individual cell size in the analyzed aggregate). If my understanding is correct, this method would reflect morphological characteristics of microfossils. Please explain whether each X/Si-O indices reflect chemical structure of microfossils or are dependent on other parameters (e.g., cell diameter of fossils).

(3) Reference for discrimination between prokaryotes and eukaryotes (Line 114-125)

Although the authors mentioned "The chemometric ratio $R_{3/2}$, i.e., the ratio of the peak intensities corresponding to CH_3 and CH_2 , has been proposed as a tool to discriminate between eukaryotic and prokaryotic biomass,,,,, than those of the latter", this seems to be incorrect. The chemometric ratio $R_{3/2}$, has been proposed as a domain-specific indicator of bacteria, archaea and possibly eukaryotes (Igisu et al., 2009). Later, Igisu et al. (2012, Environmental Microbiology Reports, 4, 42-49) reevaluated identification among three domains (bacteria, archaea, and eukaryote) and mentioned the

difficulty in discriminating three domains in natural microbial samples on the basis of only the R3/2 value. Alternatively, Qu et al. (2015) found positive correlation between the R3/2 and a Raman spectral parameter (I-1350/1600: the degree of structural order) in both eukaryotic plant fossils and cyanobacteria derived CM. The eukaryotic plant fossils were obtained from the Rhynie chert. This correlation between the R3/2 and the I-1350/1600 has been used as a tool to discriminate between eukaryotic and prokaryotic fossils (e.g., Bonneville et al., 2020 Science Advances). Therefore, I recommend that the authors compare their data with the data from Qu et al. (2015).

Minor comments:

Line 85 and later: Please make “-1” in cm⁻¹ a superscript.

Line 91: Please explain what “N-moieties” means.

Line 104-: Please explain the reason of “missorting two eukaryotes”.

Line 126-136: Please explain why the authors used the index of ester/CH₂. As the authors described, ester/CH₂ has been proposed as domain specific indicator of bacteria and archaea. On the other hand, the main object of the authors’ paper is to discrimination of prokaryotes (bacteria) and eukaryotes. The target is completely different.

Line 137-147: Please explain why the authors used the indices of C=O/CH₂ and N/CH₂. Can these indices discriminate extant prokaryotes and eukaryotes? And what does “N” in N/CH₂ mean?

Line 137-138: In Supplementary figure 2, I cannot understand whether there is significant difference in N/CH₂ ratio between BA (bacteria) and some eukaryotic specimens (e.g., PL, FG, and AR).

Line 143-147: “Sugars and proteins in bacterial extracellular polymeric substances (EPS) and cell walls,,,,,, may explain the high absorption from ester C=O in the bacterial spectra (Figure 2B).” In ester/CH₂ diagram of Supplementary Figure 2, ester/CH₂ for BA specimens is relatively lower than those for eukaryotic specimens. Please explain this discrepancy.

Line 148-163: The authors did not show raw Raman spectral data. For better understand of analytical procedure, it would be useful to add raw Raman spectral data into Supplementary material.

Line 160-161: “The position of Palaeoleptochlamys hassii suggests a composition enriched in amine-containing fossilization products.” Which part in Figure 3 should I see? Please add an explanation into the text or Figure 3.

Line 163-168: “For the enigmatic tubular nematophyte, Nematoplexus,,,,, excluding the possibility its composition was chitinous.” Possibly Figure 3 would have been obtained from various fossil samples with different preservation conditions (host rock, temperature, pressure, age and so on). I would like to see whether the data depend on preservation conditions. In addition, if possible, addition of data for cyanobacteria to Figure 3 would be helpful for the authors’ claim: the analyzed nematophyte is less cyanobacterial origin.

Line 175-179: Please add the Raman data into Supplementary material. As the authors have already performed Raman analysis, they can compare their results with those of Qu et al. (2015) while some correction (ATR vs transmission IR) will be needed.

Line 372-374 (figure 3): Please add explanation for colors and solid lines.

Supplementary material –

Supplementary material Line 52-54: Please add a figure of distribution of calcium on the sample surface. It could strengthen the authors’ interpretation on the assignment of bands at 1540 and 1575 cm⁻¹.

Supplementary Figure 5: I failed to understand how to read this figure. Which part represents polysaccharide and protein? Please add some explanation(s) in the figure.

I hope these comments will be helpful.

Reviewer #2:
Remarks to the Author:
Summary

The paper 'Molecular fingerprints resolve affinities of early terrestrial fossils' by Loron et al. uses ATR FT-IR compositional fingerprints of a diverse organic microfossil assemblage from the Devonian Rhynie Chert to quantitatively distinguish different clades of organisms. A ChemoSpace PCA of the spectroscopic fingerprints achieves a clear separation of eukaryotes and bacteria, and a discriminant analysis partially separates clusters of even more derived groups. The authors demonstrate that the Rhynie chert preserves not only exceptional morphology, but also phylogenetically informative fossilization products of original biomolecules.

Significance and context

This is an incredibly important and timely paper: the foundational observation of biological signatures in complex organic matter, published in 2020 (Wiemann et al. *Science Advances*), has been the target of criticism (Alleon et al. 2021), claiming that the spectral data collected were purely artefactual, and that no biological signatures exist in fossil organic matter, even more so, that fossil organic matter is not related to original tissue constituents. While authors have subsequently reproduced spectral signatures of individual metazoan fossils with Raman and FT-IR, biological signatures in fossil organic matter are still considered controversial. This paper by Loron et al. has the potential to be the turning point of research on biosignatures in organic fossils: it uses a complementary spectroscopy approach (FT-IR) and yet identifies the same tissue type and phylogenetic signals recovered in the 2020 study, for a completely different set of very ancient fossils. The discovery of biological signatures in fossil organic matter, and the linkage of carbonaceous films and original biomolecules, represents the greatest opportunity in the current geosciences – Loron et al. demonstrate the presence of biological signals preserved in organic fossils from a very ancient chert system, and offer a transformative application to the identification of microfossils with ambiguous morphological features. The authors focus on a more inclusive taxon-sampling, and expand our understanding of preserved biosignatures to the Biotan clade.

Working at the interface of biology, chemistry, and geology, I strongly recommend publication. This study has implications for the fields of paleontology, geochemistry, geobiology, microbiology, material sciences, and evolutionary biology: due to the general scope, *Nature Communications* is a suitable journal for this study.

Robustness and Reproducibility

Loron et al. sampled a total of $n=49$ carbonaceous microfossils using ATR FT-IR, and analyzed data via multivariate statistical analysis and machine learning. Spectral band assignments are conservative and valid. ChemoSpace clustering is exceptionally clean for spectra collected from Devonian carbonaceous fossils. K-clustering results match the ChemoSpace clusters. The assessment of R3/2 for the distinction of prokaryotes and eukaryotes is refreshing. I consider the phylogenetically meaningful separation of samples based on their vibrational properties convincing.

The discriminant spectral features correspond to those recovered in independent studies (Wiemann et al. 2020, McCoy et al. 2020). Relevant literature is cited.

Sufficient methodological detail is provided to replicate the experimental set-up, and the Supplementary Information contains helpful and relevant details and data.

Important issues to be addressed

-Endogeneity assessment: Please take a couple of spectra of the chert surrounding the organic microfossils, and run a ChemoSpace PCA of all fossil and chert spectra. I would expect at least a

partial separation of fossil and chert clusters. This should be sufficient to demonstrate endogeneity of the organic matter associated with fossils.

-Title: I think the title needs to state that you did not sample across different sites, but focused on the Rhynie chert specifically. You mention in your introduction how important it is to ground-truth biological signatures in organic matter for one specific site – I agree that elimination of different ‘taphonomic signals’ certainly helps in detecting high-quality biosignatures; but this sampling limitation needs to be clear to the reader (especially since a different study demonstrated already the presence of phylogenetic signals in Cambrian-to-Recent carbonaceous fossils from around the globe). When I initially read the title, I associated ‘terrestrial’ with ‘Earth-bound’ – you may want to avoid confusion. Please point out that you have a very inclusive taxon sampling which is certainly a strength of your study.

Minor issues to be addressed

-Fig. 2B: instead of numbering the bands, please label the functional groups to help potential readers.

-Fig. 3: first of all - it’s great to see that the fungal cluster overlaps (as to be expected based on the chitin/polysaccharide-rich composition) with the plant and arthropod material! Instead of labelling the wavenumbers, you may want to list the corresponding functional groups.

-Fig. 3: the caption mentions a ‘linear discriminant analysis’, but you plot has two axes, and cannot be a linear discriminant analysis. I believe you may have performed a Canonical Correspondence Analysis in PAST – a different type of discriminant analysis. Please make sure to correct the caption.

-Line 404: you are listing ‘Stavisky-Golay’, but you mean ‘Savitzky-Golay’. Please correct the spelling.

Reviewer #3:

Remarks to the Author:

In this manuscript, the use of Attenuated Total Reflectance Fourier Transform Infrared (ATR-FTIR) micro-spectroscopy to analyze organic matter referred to as Rhynie cherts have been reported. This technique is highly applicable to wide research areas, but its ability to support palaeontologic analysis is somehow innovative and useful, especially for any unidentified fossils. Multivariate analysis and machine-learning approaches are good complements to proceed with the ATR-FTIR spectra for further comparison. Overall, the results and scientific rationale sound well, including the convincing supplementary data. Thus, the reviewer would like to recommend this work become accepted in its present form.

REVIEWER COMMENTS

Reviewer #1 (Remarks to the Author):

This paper presents results of ATR FTIR microspectroscopy performed on organic microfossils preserved in Lower Devonian Rhynie Chert in order to investigate the relationships between the phylogenetic affinity and microspectroscopic fingerprints of microfossils within a single assemblage. Evaluating affinities of fossil microorganisms is very important for elucidation of their origin, but only morphological characteristics is often insufficient. While chemical analysis of microfossils is promising to provide broad phylogenetic information, understand of taxon-specific chemical signals in microfossils is still limited. The authors analyzed 49 individual fossils (morphologically different 13 taxa) by ATR-FTIR and Raman microspectroscopies together with statistical analysis. Finally, the authors revealed that ATR-FTIR with statistical analysis can discriminate the prokaryotic (cyanobacterial) and eukaryotic fossils.

We would like to point out, here and below, that Raman was not performed in this study. We think this reviewer may have misinterpreted our remark on line 183 about adapting methods previously used for Raman data, so we have revised the line to make it more explicit that this study adapted them for FTIR.

The present ATR-FTIR work of organic microfossils is very interesting. The authors have made a good attempt at analyzing microfossils by using ATR-FTIR. Because ATR-FTIR microspectroscopy can be applied to standard petrographic thin sections (removing from a glass slide is not needed), this can be widely applicable to geological samples. Microspectroscopy with statistical analysis helps to understand slight differences in obtained spectra. ATR-FTIR microspectroscopy together with statistical analysis could provide important insights for evaluating taxon-specific chemical signals in microfossils, however, still have some problems as indicated below.

Major comments:

(1) Peak assignments of IR spectra (Supplementary material Line 52-54)

I am wondering about the peak assignments for the analyzed samples. As far as I can see the raw spectra (Supplementary figure 3), the obtained peaks (the authors assigned as organic bands) below the 1900 cm⁻¹ region are very small. Therefore, it is questionable whether the observed peaks in figure 2 and supplementary figures is due to fossil organic matter, due to matrix, or due to surface or experimental contamination.

More careful discussion would be needed for the peak assignments for the analyzed samples. At least, an IR spectrum of chert matrix should be added in main figure or supplementary material.

We have followed these suggestions: we added more explanation of the band assignment (line 57-66 in Supplementary File) and a new spectrum from the matrix in Supplementary Figure 3. The observed peaks are consistently present at wavenumbers assignable to known organic

moieties and the PCA loadings (Figure S5) confirms that the organic bands and not the Si-O bands are responsible for discriminating the fossils. Following reviewer 2's suggestion, we ran a PCA with matrix spectra as well as the fossils (Supplementary Fig 8). As expected, matrix separates clearly from fossil material on PC1. This demonstrates the endogeneity of our organic signals.

(2) A meaning of a ratio X/silica

In supplementary material line 73-75, the authors mentioned the spectra were normalized by the 1615 cm⁻¹ silica band. This corresponds to a ratio X/Silica. In this case, CH/Si-O and other ratios depend on concentration/amount of organic matter in analyzed area. In many organic microfossils, they have hollow interior (less organic matter) and we can recognize them by wall/membrane structures. When such microfossil cell aggregates (e.g., Figure1) were analyzed by FTIR with 100 x 100um² aperture, the aggregates with smaller cells (i.e. prokaryotes) would be expected to show larger CH/Si-O than those with larger cells (i.e. eukaryotes). Therefore, I am wondering whether the PC1 in Figure2 depends on morphological parameter (e.g. individual cell size in the analyzed aggregate). If my understanding is correct, this method would reflect morphological characteristics of microfossils. Please explain whether each X/Si-O indices reflect chemical structure of microfossils or are dependent on other parameters (e.g., cell diameter of fossils).

The reviewer raises an important concern about normalization, and we have made revisions to be as clear as possible about this. Usually, the purpose of normalization in spectroscopy is to remove influence of size, thickness and concentration (Trevisan et al., 2012). In the present case the organic matter of interest is embedded in a medium (silica) which produces larger peaks than the organic matter itself, and the silica and organic peaks cannot be entirely separated. Whether we normalise on an organic peak or the large silica peak, the relative size of peaks after normalization will therefore be scaled in part by the silica/organic ratio.

The key question is whether this unavoidable confounding factor actually explains our results — in other words, whether the simple min-max normalization we performed on the 1615 cm⁻¹ silica band caused the data to separate according to the organic/silica ratio rather than the organic composition. Fortunately, we can test this, and the answer is no. In relation to the k-mean clustering, it is clear from Figure 2B that the two clusters do not differ primarily in the strength of the organic signal but in its composition: after normalizing for silica, C1 is lower in CH but higher in several organic bands than C2. Moreover, C2, which is higher in CH, consists mostly of the cyanobacteria; intuition suggests that these would preserve less organic matter than plants, arthropods, etc., not more. Similarly, in relation to the PCA results, CH/Si-O increases along PC1 — which separates prokaryotes and eukaryotes — but C=O/Si-O does not (Figures S6 and S7). If PC1 was simply reflecting the amount of organic matter in the spot, the ratios of all organic bands to silica bands should increase along it (and again, one would expect the positions of cyanobacteria and the eukaryotes to be reversed).

To make these points clearer we have revised the section of the supplement on lines 113-135 and the accompanying figure captions.

(3) Reference for discrimination between prokaryotes and eukaryotes (Line 114-125) Although the authors mentioned “The chemometric ratio $R_{3/2}$, i.e., the ratio of the peak intensities corresponding to CH_3 and CH_2 , has been proposed as a tool to discriminate between eukaryotic and prokaryotic biomass,,,,, than those of the latter”, this seems to be incorrect. The chemometric ratio $R_{3/2}$, has been proposed as a domain-specific indicator of bacteria, archaea and possibly eukaryotes (Igisu et al., 2009). Later, Igisu et al. (2012, Environmental Microbiology Reports, 4, 42-49) reevaluated identification among three domains (bacteria, archaea, and eukaryote) and mentioned the difficulty in discriminating three domains in natural microbial samples on the basis of only the $R_{3/2}$ value. Alternatively, Qu et al. (2015) found positive correlation between the $R_{3/2}$ and a Raman spectral parameter (I_{1350}/I_{1600} : the degree of structural order) in both eukaryotic plant fossils and cyanobacteria derived CM. The eukaryotic plant fossils were obtained from the Rhynie chert. This correlation between the $R_{3/2}$ and I_{1350}/I_{1600} has been used as a tool to discriminate between eukaryotic and prokaryotic fossils (e.g., Bonneville et al., 2020 Science Advances). Therefore, I recommend that the authors compare their data with the data from Qu et al. (2015).

We agree with the reviewer that clarifications and corrections were needed here. On lines 121-125 we have therefore clarified the text as follow:

“The chemometric ratio $R_{3/2}$, i.e., the ratio of the peak intensities corresponding to CH_3 and CH_2 , has been proposed as a domain indicator of bacteria, archaea and possibly eukaryotes³ because membrane lipids in archaea are characterised by shorter and more highly branched aliphatic chains than those of bacteria and eukaryotes^{3; 19-20}. However, discriminating the three domains in natural assemblages based only on this ratio may be difficult³⁴.” **Reference 34 corresponding to Igisu et al. (2012)**

We have also added a sentence on raman/FTIR correlations line 134-136: “Alternatively, in another Rhynie Chert study, prokaryotes and eukaryotes would show different trend when Raman I_{1350}/I_{1600} ratio (degree of structural order) is plotted against $R_{3/2}$ ratio, suggesting an alternate tool to discriminate domains²⁷.”

Minor comments:

Line 85 and later: Please make “-1” in cm^{-1} a superscript.

This has been corrected

Line 91: Please explain what “N-moieties” means.

This has been corrected by “Nitrogen-moieties”

Line 104-: Please explain the reason of “missorting two eukaryotes”.

K-means clustering naturally sorted eukaryote and prokaryote with 96% accuracy: 2 samples out of the 49 were missampled. We have modified this sentence accordingly (line 109) :

“K-mean clustering suggested two clusters corresponding to a similar sorting to PC1 with 96% accuracy (only 2 eukaryotes (number 26 and 35) were sorted as prokaryotes; Figure 2A, 2B)”

Line 126-136: Please explain why the authors used the index of ester/CH₂. As the authors described, ester/CH₂ has been proposed as domain specific indicator of bacteria and archaea. On the other hand, the main object of the authors' paper is to discrimination of prokaryotes (bacteria) and eukaryotes. The target is completely different.

The reviewer is correct about what has been proposed previously, but we wanted to test whether this descriptor could also sort eukaryote and prokaryotes given differences in ester contribution. As indicated on line 147, “T-tests showed significant differences in ester/CH₂ ratio between prokaryotes and eukaryotes from Rhynie, supporting its potential usefulness to discriminate domains (p value = 0.0257)”. We have clarified the justification for trying this (line 148-149).

Line 137-147: Please explain why the authors used the indices of C=O/CH₂ and N/CH₂. Can these indices discriminate extant prokaryotes and eukaryotes? And what does “N” in N/CH₂ mean?

These indices are other possible chemometric parameters, reflecting the respective contribution of carbonyl and nitrogen moieties, both being important contributors in kerogen and, furthermore in the fossilization products of sugars and proteins. N stands for nitrogen. We have completed this part to explain why these ratios are tested (line 155-158): “We tested alternative ratios, C=O/CH₂ (carbonyl/CH₂) and N/CH₂ (Nitrogen-moieties/CH₂), corresponding to the variation in contribution of products from the fossilization of sugars and proteins. These ratios differentiate the two domains in Rhynie material (T-test, p value= 0.0001 (C=O/CH₂) and 0.0071 (N/CH₂))”

Line 137-138: In Supplementary figure2, I cannot understand whether there is significant difference in N/CH₂ ratio between BA (bacteria) and some eukaryotic specimens (e.g., PL, FG, and AR).

There is a significant difference between eukaryotes as a whole and prokaryotes. Additional T-tests conducted to test the difference between PL, FG, AR and BA are giving p value of 0.0458 for PL vs BA; p= 0.0186 for FG vs BA; and p=0.0276 for AR vs BA. They are therefore significantly different. We have amended this part (line 158-161) by adding these new calculations:

“Although plants, fungus and arthropods values for N/CH₂ ratio (PL, FG, AR in Supplementary Figure 2) span a range that slightly overlap with bacteria (BA), their respective t-tests show significant differences (p value = 0.0458 (PL vs BA); 0.0186 (FG vs BA); 0.0276 (AR vs BA)).“

Line 143-147: “Sugars and proteins in bacterial extracellular polymeric substances

(EPS) and cell walls,,,,, may explain the high absorption from ester C=O in the bacterial spectra (Figure 2B).” In ester/CH₂ diagram of Supplementary Figure 2, ester/CH₂ for BA specimens is relatively lower than those for eukaryotic specimens. Please explain this discrepancy.

There is no discrepancy, bacteria show a higher absorption for ester but also a higher absorption for aliphatic moieties (including CH₂), therefore their ratio can be lower.

Line 148-163: The authors did not show raw Raman spectral data. For better understand of analytical procedure, it would be useful to add raw Raman spectral data into Supplementary material.

Raman was not performed in this study. We think the reviewer may have misinterpreted our remark on line 182 about adapting methods previously used for Raman data, so we have revised the line to make it more explicit that this study adapted them for FTIR.

Line 160-161: “The position of *Palaeolectochlamys hassii* suggests a composition enriched in amine-containing fossilization products.” Which part in Figure 3 should I see? Please add an explanation into the text or Figure 3.

We agree with the reviewer and have modified the figure with a legend.

Line 163-168: “For the enigmatic tubular nematophyte, *Nematoplexus*,,,,, excluding the possibility its composition was chitinous.” Possibly Figure 3 would have been obtained from various fossil samples with different preservation conditions (host rock, temperature, pressure, age and so on). I would like to see whether the data depend on preservation conditions. In addition, if possible, addition of data for cyanobacteria to Figure3 would be helpful for the authors’ claim: the analyzed nematophyte is less cyanobacterial origin.

The samples are all from the same Rhynie Chert blocks and have experienced the same temperature, pressure, age etc. Nematophytes were already characterised as a eukaryote by PCA, k-mean clustering and machine learning approaches using three different models, clearly distinguishing them from cyanobacteria. Figure 3 presents an analysis of eukaryotes alone to see whether they can be differentiated on the basis of their amino-glucan/non amino-glucan composition. Adding bacteria to this analysis would make it less valid as a test of this specific hypothesis, so we have not followed this suggestion.

Line 175-179: Please add the Raman data into Supplementary material. As the authors have already performed Raman analysis, they can compare their results with those of Qu et al. (2015) while some correction (ATR vs transmission IR) will be needed.

No Raman spectroscopy was performed in this study. We have revised the line 182 to make it more explicit that this study used FTIR, not Raman.

Line 372-374 (figure 3): Please add explanation for colors and solid lines.

We have added a legend on the figure as well as simplified the main text figure for an arrow (original figure is included in the supplementary).

Supplementary material –

Supplementary material Line 52-54: Please add a figure of distribution of calcium on the sample surface. It could strengthen the authors' interpretation on the assignment of bands at 1540 and 1575 cm⁻¹.

Carboxylate at these wavenumbers is expected both as a normal feature in kerogen and product of fossilization of sugars and proteins. Calcium bonded carboxylate is known to have this duet, hence our suggestion. Measuring calcium distribution on these samples at the required level of sensitivity (e.g. with EPMA) would require repolishing and carbon coating which would damage the sections and compromise future investigations and is also not possible for museum specimens.

We have reformulated this text (line 61-67) to clarify this point:

“In addition, bands at 1575 and 1540 cm⁻¹ indicate also a contribution of carboxylate groups (COO⁻), a common function in degrading organic material (Painter et al., 1981) and in fossilization products of polysaccharide rich tissues at the silica interface (Lu and Miller, 2002; Janssen et al., 2021). Interactions taking place at the silica interface will also produce ester-SiO moieties in kerogen rich in carboxylate moieties (Hantal et al., 2014).”

Supplementary Figure5: I failed to understand how to read this figure. Which part represents polysaccharide and protein? Please add some explanation(s) in the figure.

This figure represents the spectral loadings of our PCA, i.e. the discriminant variables for the clustering (here they are wavenumber as we analysed spectral data). We have revised the figure by labelling the bands to address the reviewer's comment. Please note that we are not talking of polysaccharides and proteins per se but of fossilization products from polysaccharide and protein.

I hope these comments will be helpful.

Indeed these were very helpful comments. Our sincere thanks.

Reviewer #2 (Remarks to the Author):

Summary

The paper 'Molecular fingerprints resolve affinities of early terrestrial fossils' by Loron et al. uses ATR FT-IR compositional fingerprints of a diverse organic microfossil assemblage from the Devonian Rhynie Chert to quantitatively distinguish different clades of organisms. A ChemoSpace PCA of the spectroscopic fingerprints achieves a clear separation of eukaryotes and bacteria, and a discriminant analysis partially

separates clusters of even more derived groups. The authors demonstrate that the Rhynie chert preserves not only exceptional morphology, but also phylogenetically informative fossilization products of original biomolecules.

Significance and context

This is an incredibly important and timely paper: the foundational observation of biological signatures in complex organic matter, published in 2020 (Wiemann et al. *Science Advances*), has been the target of criticism (Alleon et al. 2021), claiming that the spectral data collected were purely artefactual, and that no biological signatures exist in fossil organic matter, even more so, that fossil organic matter is not related to original tissue constituents. While authors have subsequently reproduced spectral signatures of individual metazoan fossils with Raman and FT-IR, biological signatures in fossil organic matter are still considered controversial. This paper by Loron et al. has the potential to be the turning point of research on biosignatures in organic fossils: it uses a complementary spectroscopy approach (FT-IR) and yet identifies the same tissue type and phylogenetic signals recovered in the 2020 study, for a completely different set of very ancient fossils. The discovery of biological signatures in fossil organic matter, and the linkage of carbonaceous films and original biomolecules, represents the greatest opportunity in the current geosciences – Loron et al. demonstrate the presence of biological signals preserved in organic fossils from a very ancient chert system, and offer a transformative application to the identification of microfossils with ambiguous morphological features. The authors focus on a more inclusive taxon-sampling, and expand our understanding of preserved biosignatures to the Biotan clade.

Working at the interface of biology, chemistry, and geology, I strongly recommend publication. This study has implications for the fields of paleontology, geochemistry, geobiology, microbiology, material sciences, and evolutionary biology: due to the general scope, *Nature Communications* is a suitable journal for this study.

Robustness and Reproducibility

Loron et al. sampled a total of $n=49$ carbonaceous microfossils using ATR FT-IR, and analyzed data via multivariate statistical analysis and machine learning. Spectral band assignments are conservative and valid. ChemoSpace clustering is exceptionally clean for spectra collected from Devonian carbonaceous fossils. K-clustering results match the ChemoSpace clusters. The assessment of $R_{3/2}$ for the distinction of prokaryotes and eukaryotes is refreshing. I consider the phylogenetically meaningful separation of samples based on their vibrational properties convincing.

The discriminant spectral features correspond to those recovered in independent studies (Wiemann et al. 2020, McCoy et al. 2020). Relevant literature is cited. Sufficient methodological detail is provided to replicate the experimental set-up, and the Supplementary Information contains helpful and relevant details and data.

Important issues to be addressed

-Endogeneity assessment: Please take a couple of spectra of the chert surrounding the organic microfossils, and run a ChemoSpace PCA of all fossil and chert spectra.

I would expect at least a partial separation of fossil and chert clusters. This should be sufficient to demonstrate endogeneity of the organic matter associated with fossils.

Thank you for your suggestion. A Chemospace including matrix points has been added in the supplementary.

-Title: I think the title needs to state that you did not sample across different sites, but focused on the Rhynie chert specifically. You mention in your introduction how important it is to ground-truth biological signatures in organic matter for one specific site – I agree that elimination of different ‘taphonomic signals’ certainly helps in detecting high-quality biosignatures; but this sampling limitation needs to be clear to the reader (especially since a different study demonstrated already the presence of phylogenetic signals in Cambrian-to-Recent carbonaceous fossils from around the globe). When I initially read the title, I associated ‘terrestrial’ with ‘Earth-bound’ – you may want to avoid confusion. Please point out that you have a very inclusive taxon sampling which is certainly a strength of your study.

This has been addressed, thank you very much these excellent suggestions. The new title is : “Molecular fingerprints resolve affinities of Rhynie Chert organic fossils”

Minor issues to be addressed

-Fig. 2B: instead of numbering the bands, please label the functional groups to help potential readers.

We have followed this suggestion and labelled the functional groups.

-Fig. 3: first of all - it's great to see that the fungal cluster overlaps (as to be expected based on the chitin/polysaccharide-rich composition) with the plant and arthropod material! Instead of labelling the wavenumbers, you may want to list the corresponding functional groups.

To bring more clarity to the figure (each wavenumber corresponds to different functional groups, making the figure unreadable if we labelled them all), we have revised the figure by summarizing the different wavenumbers with just an arrow. The original figure with the number corresponding to each wavenumber from Supplementary Table 2 has been added in the supplementary.

-Fig. 3: the caption mentions a ‘linear discriminant analysis’, but your plot has two axes, and cannot be a linear discriminant analysis. I believe you may have performed a Canonical Correspondence Analysis in PAST – a different type of discriminant analysis. Please make sure to correct the caption.

Thank you for pointing this out — we have now corrected it.

-Line 404: you are listing ‘Stavisky-Golay’, but you mean ‘Savitzky–Golay’. Please correct the spelling.

Indeed, thank you, this has been corrected.

Reviewer #3 (Remarks to the Author):

In this manuscript, the use of Attenuated Total Reflectance Fourier Transform Infrared (ATR-FTIR) micro-spectroscopy to analyze organic matter referred to as Rhynie cherts have been reported. This technique is highly applicable to wide research areas, but its ability to support palaeontologic analysis is somehow innovative and useful, especially for any unidentified fossils. Multivariate analysis and machine-learning approaches are good complements to proceed with the ATR-FTIR spectra for further comparison. Overall, the results and scientific rationale sound well, including the convincing supplementary data. Thus, the reviewer would like to recommend this work become accepted in its present form.

We thank the reviewer for this endorsement of our study.

Reviewers' Comments:

Reviewer #1:

Remarks to the Author:

The manuscript appears to have been improved. I would like to thank the authors for having addressed most of my comments, and sorry that I misunderstood the previous line 183-. I only have some comments on the revised manuscript. I hope that my individual comments below would be helpful for the authors' revisions. I would recommend the paper is acceptable after minor revision.

Revised line 131-

The authors conducted T-test for examination of difference in semi-quantitative ratios between prokaryotic and eukaryotic fossils. T-test is used to compare the averages of two groups and requires the assumptions which include normality of data distribution, homogeneity of variance and so on (e.g., Kim, 2015. Korean J Anesthesiol. 68, 540). I would like to suggest that the authors add the averages of semi-quantitative ratios for each sample in Supplementary Information Figure2 or its caption and mention the data of fossil materials satisfy the assumptions. Although I'm not expert in statistical analysis, there seems to be a difference in the averages of ester/CH₂, C=O/CH₂ and N/CH₂ between bacteria and eukaryotes as far as I can see Supplementary Information Figure2. However, because there are the large variations in ester/CH₂, C=O/CH₂ and N/CH₂ ratios, it is questionable whether these ratios are useful to discriminate fossil domains. It is easier for me to accept that comparing fossils by combination of these ratios (ester/CH₂, C=O/CH₂ and N/CH₂) will be useful to narrow down the origins of fossil materials.

Revised Line 139

Probably "Chert" or "material" should be inserted after Rhynie.

Revised Line 254

There is not Supplementary Information section 3d. The explanation for normalization is shown in Supplementary Information section 4b instead of 3d, isn't it?

Supplementary information Figure 1

Thank you for adding data for the chert matrix. The revised Figure2 and Supplementary Information Figures3 and 8 are very helpful for demonstrating the endogeneity of Rhynie materials. I agree that FTIR data for fossil samples is different from that for the chert matrix. However, I'm still wondering about assignment of 1615cm⁻¹ band. The 1615cm⁻¹ band is clearly observed in fossil samples but little in the chert matrix (Supplementary Information Figures3). Is there a possibility that organic component (e.g., aromatic C=C) is contributed to the 1615cm⁻¹ band instead of Si-O bond?

Supplementary information Figure 2

Please add explanations what blue boxes, black circles, black bars represent.

Supplementary information Figure 8

I think that PC1 vs PC2 score plot for fossil material in Supplementary Information Figure 8 is same data as those in Figure 2 and Supplementary Information Figures 6 and 7. If that is correct, why the distribution for fossil samples is different between Supplementary Information Figure 8 and the others?

Reviewer #2:

Remarks to the Author:

I am here reviewing the revised manuscript 'Molecular fingerprints resolve affinities of Rhynie Chert organic fossils' by Loron and colleagues. The authors have added matrix fingerprints in the SI to address my suggestion regarding a potential assessment of organic matter endogeneity, and modified

the title of their paper.

Most minor concerns and suggestions were addressed – there is still a typo in line 256 (Savitzky-Golay), please make sure to correct this before publication. I have looked at the comments by the other two reviewers and appreciate the addition of new data. I recommend the paper for publication.

REVIEWER COMMENTS

Reviewer #1 (Remarks to the Author):

The manuscript appears to have been improved. I would like to thank the authors for having addressed most of my comments, and sorry that I misunderstood the previous line 183-. I only have some comments on the revised manuscript. I hope that my individual comments bellow would be helpful for the authors' revisions. I would recommend the paper is acceptable after minor revision.

Revised line 131-

The authors conducted T-test for examination of difference in semi-quantitative ratios between prokaryotic and eukaryotic fossils. T-test is used to compare the averages of two groups and requires the assumptions which include normality of data distribution, homogeneity of variance and so on (e.g., Kim, 2015. Korean J Anesthesiol. 68, 540). I would like to suggest that the authors add the averages of semi-quantitative ratios for each sample in Supplementary Information Figure2 or its caption and mention the data of fossil materials satisfy the assumptions.

Indeed, after inspection some of the data do not follow a normal distribution. We therefore switched from the T-test to nonparametric Mann-Whitney *U* tests (after consulting with a statistician). These tests continue to show significant differences between prokaryotes and eukaryotes ($p = 0.02$) so our main conclusion is unaltered. We clarified the reason for the choice of test on lines 284-291.

We added the mean value of each group for each ratio on Supplementary Figure 2.

Although I'm not expert in statistical analysis, there seems to be a difference in the averages of ester/CH₂, C=O/CH₂ and N/CH₂ between bacteria and eukaryotes as far as I can see Supplementary Information Figure2. However, because there are the large variations in ester/CH₂, C=O/CH₂ and N/CH₂ ratios, it is questionable whether these ratios are useful to discriminate fossil domains. It is easier for me to accept that comparing fossils by combination of these ratios (ester/CH₂, C=O/CH₂ and N/CH₂) will be useful to narrow down the origins of fossil materials.

We agree with the reviewer (and editor). The comparisons made between bacteria and individual eukaryotic groups (formerly on lines 141 and 149) were of low statistical power so we have removed them. We also added the following points in agreement with the reviewer, on lines 143-145 and lines 175-176: *"However, the large variation between eukaryote groups (Supplementary Fig. 2) suggests that the ester/CH₂ ratio alone may not reliably discriminate between fossil domains."* and *"Taken together, ester/CH₂, C=O/CH₂ and N/CH₂ chemometric ratio constitute useful tools to narrow down the origins of fossil material in the Rhynie chert."*

Revised Line 139

Probably “Chert” or “material” should be inserted after Rhynie.

Done — thanks.

Revised Line 254

There is not Supplementary Information section 3d. The explanation for normalization is shown in Supplementary Information section 4b instead of 3d, isn't it?

Corrected — thanks.

Supplementary information Figure 1

Thank you for adding data for the chert matrix. The revised Figure 2 and Supplementary Information Figures 3 and 8 are very helpful for demonstrating the endogeneity of Rhynie materials. I agree that FTIR data for fossil samples is different from that for the chert matrix. However, I'm still wondering about assignment of 1615 cm⁻¹ band. The 1615 cm⁻¹ band is clearly observed in fossil samples but little in the chert matrix (Supplementary Information Figures 3). Is there a possibility that organic component (e.g., aromatic C=C) is contributed to the 1615 cm⁻¹ band instead of Si-O bond?

Yes, it is probable that aromatic C=C moieties contributed to the 1615 cm⁻¹ band in addition to silica. However, this would make the band hazardous to use for analyses because of this mixed signal. We don't see a clear, marked absorption for aromatic CH stretch around ca. 3100 cm⁻¹ so the contribution of such groups is probably far less than the silica. We completed the section 3b with this thought.

Supplementary information Figure 2

Please add explanations what blue boxes, black circles, black bars represent.

Done (in the caption) — thanks.

Supplementary information Figure 8

I think that PC1 vs PC2 score plot for fossil material in Supplementary Information Figure 8 is same data as those in Figure 2 and Supplementary Information Figures 6 and 7. If that is correct, why the distribution for fossil samples is different between Supplementary Information Figure 8 and the others?

This is indeed the same data. In Fig 2 and Sup fig 6 and 7, the specimens were scattered along PC1 (the maximum variance) and PC2 (the second most important variance). When introducing the matrix samples, the maximum variance for the set became the differences between matrix sample and fossil sample, hence the new distribution.

Reviewer #2 (Remarks to the Author):

I am here reviewing the revised manuscript 'Molecular fingerprints resolve affinities of Rhynie Chert organic fossils' by Loron and colleagues. The authors have added matrix fingerprints in the SI to address my suggestion regarding a potential assessment of organic matter endogeneity, and modified the title of their paper.

Most minor concerns and suggestions were addressed – there is still a typo in line 256 (Savitzky-Golay), please make sure to correct this before publication. I have looked at the comments by the other two reviewers and appreciate the addition of new data. I recommend the paper for publication.

Corrected – thanks.

Reviewers' Comments:

Reviewer #1:

Remarks to the Author:

The authors have addressed my comments. I would like to recommend the revised manuscript for publication. However, the abstract should contain no references while I did not realize it until now. Please check the abstract and reference numbers in main text before publication.

REVIEWERS' COMMENTS

Reviewer #1 (Remarks to the Author):

The authors have addressed my comments. I would like to recommend the revised manuscript for publication. However, the abstract should contain no references while I did not realize it until now. Please check the abstract and reference numbers in main text before publication.

Thanks, this has been corrected.